# StarDICE II: Calibration of an Uncooled Infrared Thermal Camera for Atmospheric Gray Extinction Characterization

**DOI:** 10.3390/s24144498

**Published:** 2024-07-11

**Authors:** Kélian Sommer, Bertrand Plez, Johann Cohen-Tanugi, Sylvie Dagoret-Campagne, Marc Moniez, Jérémy Neveu, Marc Betoule, Sébastien Bongard, Fabrice Feinstein, Laurent Le Guillou, Claire Juramy, Eduardo Sepulveda, Thierry Souverin

**Affiliations:** 1Laboratoire Univers et Particules de Montpellier, Université de Montpellier, Centre National de la Recherche Scientifique, F-34095 Montpellier, France; bertrand.plez@umontpellier.fr (B.P.); johann.cohen-tanugi@in2p3.fr (J.C.-T.); 2Laboratoire de Physique de Clermont, Université Clermont Auvergne, Centre National de la Recherche Scientifique, F-63000 Clermont-Ferrand, France; 3IJCLab, Université Paris-Saclay, Centre National de la Recherche Scientifique, F-91405 Orsay, France; 4LPNHE, Centre National de la Recherche Scientifique & Sorbonne Université, 4 Place Jussieu, F-75005 Paris, France; 5Centre National de la Recherche Scientifique, Aix-Marseille University, CPPM, 163 Avenue de Luminy, F-13009 Marseille, France

**Keywords:** uncooled infrared thermal camera calibration, cirrus cloud, atmosphere monitoring, gray extinction, low radiances

## Abstract

The StarDICE experiment strives to establish an instrumental metrology chain with a targeted accuracy of 1 mmag in *griz* bandpasses to meet the calibration requirements of next-generation cosmological surveys. Atmospheric transmission is a significant source of systematic uncertainty. We propose a solution relying on an uncooled infrared thermal camera to evaluate gray extinction variations. However, achieving accurate measurements with thermal imaging systems necessitates prior calibration due to temperature-induced effects, compromising their spatial and temporal precision. Moreover, these systems cannot provide scene radiance in physical units by default. This study introduces a new calibration process utilizing a tailored forward modeling approach. The method incorporates sensor, housing, flat-field support, and ambient temperatures, along with raw digital response, as input data. Experimental measurements were conducted inside a climatic chamber, with a FLIR Tau2 camera imaging a thermoregulated blackbody source. The results demonstrate the calibration effectiveness, achieving precise radiance measurements with a temporal pixel dispersion of 0.09 $W\;m^{-2}\;{\mathrm{sr}}^{-1}$ and residual spatial noise of 0.03 $W\;m^{-2}\;{\mathrm{sr}}^{-1}$. We emphasize that the accuracy of scene radiance retrieval can be systematically affected by the camera’s close thermal environment, especially when the ambient temperature exceeds that of the scene.

## 1. Introduction

Ground-based astronomical surveys like PanSTARRS [1], the Dark Energy Survey [2], and the Vera Rubin Observatory Legacy Survey of Space and Time (LSST) [3] demand precise broadband photometry [4]. The LSST Dark Energy Science Collaboration requires achieving 5 milli-magnitudes (mmag) and 1 mmag precision in ugrizy filters for first and 10-year type 1a supernovae (SNe Ia) analyses, respectively [5]. Meeting this standard is crucial for deriving competitive cosmological insights from SNe Ia data.

In this respect, the StarDICE (Star Direct Illumination Calibration Experiment [6]) metrology experiment has emerged with the intended purpose of measuring and comparing the brightness of stars from the CALSPEC library of spectrophotometric standards [7] against laboratory flux references, represented by photodiodes calibrated by the National Institute of Standards and Technology (NIST) [8]. The improved standards catalog provided by StarDICE will serve as the foundation for propagating calibration to stellar sources that are or will be observed by telescopes such as the Simonyi Survey Telescope of the Vera C. Rubin Observatory [9]. The StarDICE pathfinder project [10] has determined that the last major source of systematic uncertainty exceeding the mmag threshold is caused by the atmosphere. Directly measuring and correcting for atmospheric transmission variations will be necessary to significantly enhance the precision of the observations. Specifically, thin high-altitude cirrus clouds can induce spurious variations known as *gray extinction* [11,12]. They are challenging to detect but it is essential to properly characterize them to extract above-atmosphere reference star fluxes.

In the context of this astronomical calibration experiment, we explored a possible solution relying on an uncooled infrared thermal camera (UIRTC) to evaluate astronomical observation quality with respect to gray extinction contamination.

The utilization of IR thermal cameras has emerged as a useful source of information for atmospheric investigations through precise radiometric measurements [13,14,15,16,17,18]. Prior research has demonstrated these sensors’ ability to provide high-speed, compact, and cost-effective means to characterize atmospheric conditions by imaging the sky downwelling radiance in the long-wave infrared (LWIR) range from 8 to 14 µm, centered around the atmosphere transparency window in the 10–12 µm range [19,20]. In this spectral region, after accounting for the effect of water vapor, the contrast between cloud radiance and clear sky background radiance is relatively high, exceeding a difference of 0.1 $W\;m^{-2}\;{\mathrm{sr}}^{-1}$ [19]. Previous work established the feasibility of using UIRTCs to obtain reliable radiometric atmosphere measurements for astronomy-related applications [21,22]. Klebe et al. [17] already observed the direct sky’s thermal radiance by developing an all-sky IR thermal camera instrument. They successfully determined the sky cover fraction and estimated the precipitable water vapor (PWV), which impacts near-IR photometry. Hack et al. [23] advanced this work by comparing measurements and simulations of sky downwelling radiance. Under stable atmospheric conditions, they achieved PWV determination with an accuracy of approximately 2%. This required an instrument that had undergone prior radiometric calibration.

For conventional cooled thermal cameras, the radiometric calibration model is formulated by building the relation between image pixel values and corresponding blackbody temperatures in a controlled laboratory setting [24]. This model is presumed to be long-term effective in real-world applications as the camera will operate at the same stabilized temperature. This assumption is unsuitable for uncooled thermal cameras for which temporal and spatial non-uniformities prevail on images, leading to substantial errors in radiance retrieval when the camera operates in unstable conditions, such as varying ambient temperatures [14,19,25]. Therefore, the use of UIRTCs presents challenges that must be overcome before they can be used for atmosphere monitoring purposes as Ribeiro-Gomes et al. [26] pointed out, including (i) spatial non-uniformity; (ii) ambient and sensor temperature drift while the object’s temperature remains constant, due to uncompensated focal plane array (FPA) temperature variations; (iii) the FPA temperature-dependent response; and (iv) fixed-pattern noise (FPN).

To address the aforementioned challenges, the work presented here introduces a shutter-based radiometric calibration method designed specifically for UIRTCs and optimized for low-temperature measurements. Our method relies on a model that integrates camera housing, FPA, flat-field support and ambient temperature variations, rendering it suitable for instruments operating in less stable environmental conditions. This approach mitigates signal perturbations arising from fluctuations in ambient temperature and the internal heating of the camera. We also developed a forward model to estimate the total scene radiance in $W\;m^{-2}\;{\mathrm{sr}}^{-1}$.

This paper is structured as follows. Section 2.1 provides the rationale for this study, discussing the current state-of-the-art calibration methods. Section 2.2 describes the experimental setup. The real-time non-uniformity correction (NUC) procedure is detailed in Section 2.3. Section 2.4 introduces the scene radiance model, while Section 2.5 focuses on its specific features for laboratory measurements. The complete radiometric calibration model is presented in Section 2.7. Section 2.8 details the data acquisition procedure and dataset. The data fitting method using the least-squares algorithm is explained in Section 2.9. Section 3.1 presents the results of the improved flat-field correction method. The calibration results are analyzed in Section 3.2 and Section 3.3. Section 3.4 discusses the application of the calibration method to a real case study and compares radiometric images to radiative transfer simulations. Section 4 examines potential limitations of the calibration procedure and suggests revisions. Finally, Section 5 summarizes the main findings and outlines future work.

## 2. Materials and Methods

Our goal for calibrating a UIRTC centers on attaining the precise characterization of atmospheric gray extinction, with a specific emphasis on measuring the low radiance of the sky. This comprises an experimental setup ensuring controlled conditions, a systematic data acquisition process, and a novel calibration model for enhanced radiometric measurements. An improved in situ non-uniformity correction method mitigates potential artifacts. Performance metrics and data fitting processes validate the calibration model’s efficiency. This concise and integrated approach ensures precise thermal products suitable for atmospheric gray extinction studies.

### 2.1. Motivation

The primary objective of the calibration experiment is to correct raw observations for defects and to convert the analog-to-digital units (ADUs) read from the camera to the emitted sky radiance $W\;m^{-2}\;{\mathrm{sr}}^{-1}$, establishing a relationship with true physical radiances. Once true radiance images of the sky are obtained, cloud structures can be extracted. The conceptual principles of the calibration process are the following. Images of a blackbody emitter at multiple temperatures are taken with the camera placed in a temperature-controlled environment. A pixel response model is fitted on these images to produce calibration matrices that will be used a posteriori to calibrate in-field images. The flowchart shown in Figure 1 summarizes this process. The left block represents the laboratory calibration experiment on a reference blackbody source in controlled environmental conditions. The right block represents the application of the calibration model to raw data during operations, producing true-radiance radiometric images.

Many methods have been proposed and implemented to address this issue, specifically to correct for the different noise sources mentioned above. For a concise overview and state-of-the-art comparisons, we refer the reader to the work of González-Chávez et al. [27]. Several approaches focus on shutterless calibration methods, as relying on the shutter for flat-field support leads to interruptions and a reduction in the maximum frame rate of the cameras. In our specific use case, the flat-field-based method did not impact our research, as we leverage the overhead between CCD optical exposures (i.e., while the filter wheel is rotating or when reading the previous image). The majority of methods focus solely on temperature measurements, utilizing the temperature of the calibration blackbody source as a reference. The process typically involves a two-step approach: (i) correcting for sensor temperature drift and camera self-emission on raw digital images; (ii) performing temperature radiometric calibration by associating corrected ADUs with blackbody temperature setpoints. However, these methods rely on assumptions that are effective only under specific conditions. Notably, when working with low blackbody source temperatures (i.e., sub-zero degrees Celsius), considerations for the emissivity of the blackbody source and ambient contributions become crucial for accurately evaluating the observed radiance perceived by the camera. To correct ADUs, most techniques first assumed that the ambient contribution to the scene radiance is consistent across all camera and blackbody temperature setpoints. Images are then normalized to the user-defined reference level to stabilize the image signal. Nonetheless, in practice, temperature changes in the climate chamber housing the camera (especially when near the calibration blackbody) can cause the camera to perceive radiance originating partially from the chamber itself. The colder the blackbody source temperature, the higher the ambient scene contribution to the total scene radiance.

In this study, we undertook a different approach by developing a radiance calibration model that accounts for this effect and applied it to our dataset, adding ambient thermal emission to the blackbody source in the observed scene radiance.

### 2.2. Experimental Setup

The experimental calibration setup is depicted in Figure 2, and a simplified sketch is shown in Figure 3. The blackbody source controller stands on the center of the left image. On top sits the Raspberry Pi 4 accessed through SSH with a desktop PC on the left. Cables for the IR camera, temperature sensors, and blackbody emitting surface area pass through the climatic chamber within the intended cable sleeve. The thermal camera is positioned within a climatic chamber, serving as an environmental enclosure to replicate ambient temperatures akin to those that will be experienced during in situ measurements at the Observatoire de Haute-Provence to monitor the cloud coverage. It receives homogeneous and isotropic thermal emission from the blackbody source located at a specific fixed distance to cover the entire field-of-view (FOV) of the camera. The camera is attached to a telescope dovetail bar facing the blackbody target with the flat-field support in operation position. The temperature sensor is attached onto a non-moving plate emulating the FFC plate to prevent mechanical issues due to moving cables.

The thermal camera used for this experiment was the FLIR Tau2 (FLIR Systems Inc., Wilsonville, OR, USA) operating in the LWIR range (8–14 µm). It was coupled to a Umicore athermalized 60 mm f#1.25 lens (the f#-number is defined as the ratio of the focal length to the diameter of the aperture), which provides a scale of 1 arcmin/pixel. The sensor is an array of 640 × 512 vanadium oxide (VOx) pixels of a 17 µm square size. Figure 4 depicts the camera’s throughput as a function of wavelength, as given by the manufacturer for a typical Tau2 core model. The announced operational scene temperature range in high-gain mode is −40 °C to +160 °C. In radiometric mode, the incident radiance is digitized over a 14-bit depth at a high frame rate of 8.33 Hz provided by the ThermalGrabber USB 2.0 interface from TeAx. Two temperature sensors are implemented inside the camera. One is located on the FPA next to the microbolometers array and provides temperatures with 0.1 °C resolution. The other is built-in inside the camera’s housing (closer to the outer walls of the case) and has a resolution of 0.01 °C.

The manufacturer announces a Noise-Equivalent Temperature Difference (NETD) of 50 mK for a f#1.0 lens at a $T_{ref}$ = 300 K target temperature with unit emissivity (FLIR technical note: https://www.flir.eu/support-center/oem/what-is-the-sensitivity-of-tau-cameras-in-watts-per-m2-/, accessed on 1 March 2023). This figure of merit defines the minimum temperature difference that can be detected with a signal-to-noise ratio (SNR) of 1. Transforming this sensitivity value into the Noise-Equivalent Radiance Difference (NERD) for this specific camera lens combination requires the following calculation:(1)$$\mathrm{NERD}={f\#}^{2}\;\cdot \;\left[\int_{\lambda_{min}}^{\lambda_{max}}B\left(\lambda ,T_{ref}+\mathrm{NETD}\right)\;\cdot \;R\left(\lambda \right)\;\cdot \;d\lambda -\int_{\lambda_{min}}^{\lambda_{max}}B\left(\lambda ,T_{ref}\right)\;\cdot \;R\left(\lambda \right)\;\cdot \;d\lambda \right]$$
with B(λ,T) the Planck function and R(λ) the instrument throughput shown in Figure 4. For this camera and target temperature of 300 K, the NERD equivalent to the NETD of 50 mK is equal to 0.044 $W\;m^{-2}\;{\mathrm{sr}}^{-1}$. Figure 5 plots the NERD as a function of target temperature (blue scale) and equivalent scene radiance (red scale). The plotted curves can be interpreted as a lower boundary spatial noise limit per image.

The blackbody calibrator source is the IR-2101/301 model from Infrared Systems Development. This source provides a 63.5 × 63.5 mm^2^ emitting surface area with temperatures ranging from −30 °C to +80 °C. The blackbody emissivity is equal to 0.96±0.02, with a ±0.1 °C temperature monitoring sensor resolution calibrated with a ±0.2 °C uncertainty level relative to an NIST standard (https://www.infraredsystems.com/Products/blackbody2100.html, accessed on 1 March 2023). The operation of the blackbody is achieved through a standard RS-232 serial port. Blackbody temperatures used in the calibration process ranged from −30 °C to 60 °C, which covers the linear range of the camera response down to the lowest achievable temperature in the experimental setup conditions.

For the external flat-field support, we use a thin 2 mm copper plate placed at a ∼5 cm distance from the camera, facing the lens and covered with high-emissivity 3M Scotch Super 88 Vinyl electrical tape. Benirschke and Howard [28] and Avdelidis and Moropoulou [29] used this tape, with emissivity ϵ=0.95 at ∼50 °C. FLIR also recommends using this electrical tape and states that it has an emissivity of 0.96 in the long wavelength (8–12 µm) range (https://www.flir.com/discover/rd-science/use-low-cost-materials-to-increase-target-emissivity/, accessed on 1 March 2023). The plate is attached to a high-torque servomotor that rotates it in front of the camera lens. A Raspberry Pi controls the servomotor movement through pulse with modulation (PWM) signals. The in situ shutter-based type correction is performed at regular intervals of 30 s. This consists of positioning the flat-field support in front of the camera lens and activating the non-uniformity correction (NUC, equivalent to FFC) with the camera control-command software (FLIR Document Number 102-PS242-43 Version 133 June 2015: https://flir.netx.net/file/asset/12423/original/attachment accessed on 1 October 2022) The flat-field support surface is monitored by a temperature sensor. One measurement is sampled just before starting the NUC procedure and given to the camera microcontroller as required for internal computation [30,31].

All instruments, including the camera, the blackbody source, the external flat-field system, and temperature sensors were positioned inside a climatic chamber, which consists of an insulated facility equipped with heating and cooling systems, as well as control electronics, that are used to create a stable and repeatable environment. We used the Weisstechnik LabEvent L T/150/70/3 climatic chamber at IJCLab in Orsay. Relative humidity was kept below 2%, which was required to cool the blackbody source down to −30 °C without producing frost on the surface. The aperture window of the climatic chamber was covered with several layers of aluminum foils to prevent exterior stray light from disrupting the measurements.

### 2.3. Improved In Situ Non-Uniformity Correction

Non-uniformity correction (NUC) is essential for mitigating pixel-to-pixel sensitivity variations in thermal cameras, primarily caused by differences in the microbolometer characteristics [32]. The most common approach, known as the shutter-based flat field correction (FFC) method, relies on capturing periodic images with a closed shutter [30,31]. This process allows for precise adjustments to the pixel offsets, effectively reducing fixed-pattern noise (FPN) and ensuring the generation of accurate and uniform images. Generally, the most used FFC method triggers a NUC at regular intervals or with a $\Delta T_{FPA}$ temperature change and often falls short of achieving optimal correction. While suitable for applications with less stringent accuracy requirements, it has limitations regarding vignetting correction due to optics placed in front of the shutter. One solution, proposed by Budzier and Gerlach [33] involves the fine-tuning of the camera firmware to process and save FFC frames for post-processing. Implementing such methods is challenging due to limited access and knowledge of the embedded software governed by the manufacturer’s discretion.

Two patents have been filed by the camera manufacturer that describe the method operating internally [30,31]. In short, multiple images of the FFC support are averaged, the FPA temperature is read, and some pixel-to-pixel scaling map is computed with pre-configured tables stored in the camera firmware.

One alternative approach available to improve the process involves activating the camera’s external NUC mode. While this method employs the default correction algorithm, it uses an emitting surface positioned in front of the camera optics.

Figure 7 depicts such an implementation of the procedure for our setup. As the user can provide any type of surface in front of the lens during the FFC operation, using a homogeneous high-emissivity support can improve the correction compared to the internal shutter. In particular, it will remove any emission or vignetting due to the lens. This workaround provides a viable option when extensive modifications to the camera’s firmware are not feasible.

### 2.4. Generic Scene Radiance Model

The scene radiance observed by a detector in the general case is presented in Figure 8. The radiance perceived by the camera FPA is the sum of scene radiance passing through the atmosphere (red arrow), the background radiance reflected onto the scene passing through the atmosphere (blue arrow), the atmospheric radiance, and the ambient radiance surrounding the camera (gray arrow). This ambient radiance (green arrow) is the direct emission of the environment surrounding the camera. The contributions to the scene radiance can be summarized in the following equation:(2)$$\begin{array}{ccc} L_{scene}\left(\lambda \right) & = & \;\tau_{atm}\left(\lambda \right)\times L_{obj}\left(\lambda ,T_{obj}\right)+\tau_{atm}\left(\lambda \right)\times \left[1-\epsilon_{obj}\left(\lambda \right)\right]\times L_{bkg}\left(\lambda ,T_{bkg}\right)\\ & & +\left[1-\tau_{atm}\left(\lambda \right)\right]\times L_{atm}\left(\lambda ,T_{atm}\right)+L_{amb}\left(\lambda ,T_{amb}\right) \end{array}$$
where $L_{scene}\left(\lambda \right)$ is the sum of spectral radiances ($W\;m^{-2}\;{\mathrm{sr}}^{-1}{\mu m}^{-1}$) arriving at the detector; λ is the wavelength; $\tau_{atm}$ is the atmosphere transmission coefficient; $\epsilon_{obj}$, the object emissivity; and $L_{obj}$, $L_{bkg}$, $L_{amb}$, and $L_{atm}$ are the radiances emitted by the object, the background surrounding the scene reflected by the object, the ambient environment surrounding the camera, and the atmosphere respectively. We assume that the object reflection coefficient is $1-\epsilon_{obj}$.

UIRTCs measure the integral $L_{scene}^{inst}$ of the spectral radiance emitted by the scene within the sensor spectral band limits $\lambda_{min}\leq \lambda \leq \lambda_{max}$ and throughput R(λ):(3)$$L_{scene}^{inst}=\int_{\lambda_{min}}^{\lambda_{max}}L_{scene}\left(\lambda \right)\times R\left(\lambda \right)\times d\lambda$$
with $L_{scene}^{inst}$ being the scene radiance observed by the instrument in $W\;m^{-2}\;{\mathrm{sr}}^{-1}$. $R\left(\lambda \right)=R_{optic}\left(\lambda \right)\times R_{sensor}\left(\lambda \right)$ is the combination of optics and sensor throughputs (see Figure 4).

### 2.5. Close-Up View Radiance Model

For close-up views, such as laboratory measurements, atmospheric emission, and transmission do not affect the flux as optical thickness is negligible: $\tau_{atm}∼1$ and $L_{atm}$ = 0. It is necessary to keep the ambient radiance component for reasons explained in Section 2.7. Equation (Equation 2) simplifies to the following expression for our laboratory measurements:(4)$$L_{scene}\left(\lambda \right)=L_{obj}\left(\lambda ,T_{obj}\right)+\left[1-\epsilon_{obj}\left(\lambda \right)\right]\times L_{bkg}\left(\lambda ,T_{bkg}\right)+L_{amb}\left(\lambda ,T_{amb}\right)\;.$$

A commonly employed calibration radiation source is a blackbody emitter source with temperature regulation that has a homogeneous and isotropic emission surface with high emissivity [34]. Strictly speaking, this is a gray body with a uniform emissivity $\epsilon_{BB}$ close to 1. The radiance emitted by this blackbody at temperature $T_{BB}$ and captured by the UIRTC can be calculated using the standard Planck formula $B(\lambda ,T_{BB})$:(5)$$\epsilon_{BB}\times L_{BB}^{inst}\left(T_{BB}\right)=\epsilon_{BB}\times \int_{\lambda_{min}}^{\lambda_{max}}B\left(\lambda ,T_{BB}\right)R\left(\lambda \right)\;d\lambda =\epsilon_{BB}\times \int_{\lambda_{min}}^{\lambda_{max}}\frac{2hc^{2}}{\lambda^{5}}\frac{1}{e^{\frac{hc}{\lambda k_{B}T_{BB}}}-1}\;R\left(\lambda \right)d\lambda$$
where $L_{BB}^{inst}$ represents the radiance integrated in the bandpass of the instrument, $\epsilon_{BB}$ is the blackbody emissivity, $B(\lambda ,T_{BB})$ is the spectral radiance of a blackbody radiator at temperature $T_{BB}$, *h* is the Planck constant, *c* is the speed of light, and $k_{B}$ is the Boltzmann constant. Lastly, the integrated radiance observed by the camera during calibration is
(6)$$L_{scene}^{inst}=\epsilon_{BB}\times L_{BB}^{inst}\left(T_{BB}\right)+\left[1-\epsilon_{BB}\right]\times L_{bkg}^{inst}\left(T_{amb}\right)$$
with $L_{bkg}^{inst}$ being the background emission at ambient temperature $T_{amb}$ of the climatic chamber, and the blackbody radiance $\epsilon_{BB}\times L_{BB}^{inst}$ is the object radiance from Equation (Equation 4). The background radiance contribution is a reflection of the ambient radiance onto the blackbody surface filling the camera’s FOV.

### 2.6. Thermal Interactions inside the Camera

The lower panel of Figure 8 focuses on the heat exchanges inside the camera and derives from the model of Tempelhahn et al. [35]. $L_{scene}$ is the radiance of the scene entering the camera in the projected solid angle $\omega_{scene}$ (orange). $L_{cam}$ is the camera housing emission within the solid angle $\omega_{cam}$ (purple). $L_{pix}$ is the pixel self-emission into the half-space projected solid angle $w_{pix}$ (cyan). Tempelhahn et al. [35] proposed an expression for the net radiant flux $\phi_{i,j}$ on pixel (*i*, *j*) with the corresponding projected solid angles $\omega_{cam,i,j}$ and $\omega_{pix}$, respectively:(7)$$\phi_{i,j}=A_{pix}\times \left(\omega_{scene,i,j}\times L_{scene}^{inst}+\omega_{cam,i,j}\times L_{cam}^{sens}\left(T_{cam}\right)-\omega_{pix}\times L_{pix}^{sens}\left(T_{FPA}\right)\right)$$
where $A_{pix}$ is the pixel surface area, $L_{scene}^{inst}$ is the scene radiance defined in Equation (Equation 6), and $L_{cam}^{sens}$ and $L_{pix}^{sens}$ are the camera housing emission (computed by integrating the Planck function with camera temperature $T_{cam}$) toward the microbolometers and the pixel self-emission integrated over the sensor throughput. The emissivity of the microbolometers is assumed to be $R_{sensor}$. Tempelhahn et al. [36] have shown that the pixel’s FOV covers nearly the entire half-space for this kind of infrared thermal camera. Therefore, the corresponding pixel’s projected solid angle $\omega_{pix}$ amounts almost to π and is constant for all pixels independent of their position. The projected solid angle $\omega_{scene,i,j}$ depends on the optic focal ratio: $\omega_{scene}=\pi \sin^{2}\left[\arctan\left(1/2f\#\right)\right]=\pi /7.25$ for the lens mounted on our camera with f# = 1.25. The remaining projected solid angle $\omega_{cam,i,j}=\pi -\omega_{scene,i,j}$. The projected solid angles $\omega_{scene,i,j}$ and $\omega_{cam,i,j}$ of each pixel (i,j) depend on the position within the FPA and are symmetrically distributed around the optical axis of the IR optics.

### 2.7. Calibration Model

The radiant net flux $\phi_{i,j}$ on each pixel (i,j) induces the temperature increase in the microbolometer. The voltage *V* read by the analog-to-digital converter (ADC) and converted to analog-to-digital units (ADUs) *S* is proportional to the thermal resistance variation $\Delta R_{th}$ of the microbolometer $S\propto V=I_{bias}\times \Delta R_{th}$ [37]. In the range of temperatures considered, we assume that the resistance and temperature variations are linearly related. We discuss the non-linearity in Section 4.3. In our baseline model, we linearly express the raw response as a function of the net flux:(8)$$S_{i,j}=r_{i,j}\times \phi_{i,j}+o_{i,j}$$
where $S_{i,j}$ is the camera’s raw response of the (i, j) pixel expressed in ADUs, $r_{i,j}$ is the sensor responsivity in ADU/W, and $o_{i,j}$ is the offset signal in ADUs. Note that the gain and offset parameters are given for a unique pixel, corresponding to a unique microbolometer, as a consequence of the non-uniformity of the responses caused by optical relative irradiance, housing straylight, and detector pixel-to-pixel differences due to the manufacturing process [38].

By injecting the radiant flux expression from Equation (Equation 7) into Equation (Equation 8), we can separate the scene radiance from other contributions:(9)$$L_{obs}^{inst}=g_{i,j}\times \left(S_{i,j}-o_{i,j}\right)-L_{cam}^{sens}\times \frac{\omega_{cam,i,j}}{\omega_{scene,\;i,j}}+L_{pix}^{sens}\times \frac{\omega_{pix}}{\omega_{scene,\;i,j}}\;.$$
The pixel gain $g_{i,j}$ is expressed as
(10)$$g_{i,j}=\frac{1}{r_{i,j}A_{pix}\omega_{scene,\;i,j}}\;.$$
The individual offsets are
(11)$$M_{o,\;i,j}=-g_{i,j}\times o_{i,j}$$
(12)$$M_{cam,\;i,j}\left(T_{cam}\right)=-\frac{\omega_{cam,\;i,j}}{\omega_{scene,\;i,j}}\times L_{cam}^{sens}=-\alpha_{i,j}\times L_{cam}^{sens}$$
(13)$$M_{pix,\;i,j}\left(T_{FPA}\right)=\frac{\omega_{pix}}{\omega_{scene,\;i,j}}\times L_{pix}^{sens}=\beta_{i,j}\times L_{pix}^{sens}$$
where $M_{o,\;i,j}$ represents the offset from Equation (Equation 8), $M_{cam,\;i,j}\left(T_{cam}\right)$ represents the camera housing emission, and $M_{pix,\;i,j}\left(T_{FPA}\right)$ represents the microbolometer’s outward self-emission, with $T_{FPA}$ being the temperature of the focal plane array. $T_{cam}$ closely follows the evolution of $T_{FPA}$ due to heat propagation and equalization processes [35], except for in some instants. Therefore, it is difficult to separate the effect of the camera housing emission $M_{cam,\;i,j}\left(T_{cam}\right)$ from the pixel self-emission $M_{pix,\;i,j}\left(T_{FPA}\right)$. The sensor temperature $T_{FPA}$ is provided by the camera itself, and it is assumed that any variations are uniformly distributed across the entire sensor array. In contrast, the housing temperature sensor is placed at an unknown location inside the camera core, and the associated thermal effects are not uniform.

We found that the residuals up to −0.6 $W\;m^{-2}\;{\mathrm{sr}}^{-1}$  were correlated with the temperature difference $T_{amb}-T_{amb}\left(@FFC\right)$ (see Figure 9). Through including an additional term for the difference $L_{amb}$−$L_{amb}\left(@FFC\right)$ with the nuisance parameter γ, the residuals are reduced as shown in Figure 9. $L_{amb}$ and $L_{amb}\left(@FFC\right)$ are computed by integrating the Planck function over the total instrument throughput with the ambient temperature $T_{amb}$ at each image instant and at the FFC time. This is due to diffuse radiation entering the camera through the lens. When performing an external FFC, the camera is fed with the temperature of the external shutter $T_{FFC}$ at this specific time. However, this does not take into account ambient radiation from outside the FOV reaching the front lens of the camera and diffusing inside the housing. If the temperature of the environment changes after the FFC, this contribution is modified and affects the true scene radiance received by the FPA.

The complete calibration model is written as follows:(14)$$L_{obs}^{inst}=g_{i,j}\times \left(S_{i,j}-o_{i,j}\right)-\alpha_{i,j}\times L_{cam}^{sens}+\beta_{i,j}\times L_{pix}^{sens}+\gamma_{i,j}\times \left(L_{amb}^{inst}-L_{amb}^{inst}\left(@FFC\right)\right)\;.$$

The vector of the parameters to be adjusted for each pixel by the regression algorithm is $\theta_{i,j}=\left\{g_{i,j},\;o_{i,j},\;\alpha_{i,j},\;\beta_{i,j},\;\gamma_{i,j}\right\}$.

The temperature dependence correction and the radiometric calibration are performed simultaneously. The two-step procedure, first compensating the raw signal $S_{i,j}$ for ambient temperature changes and then transforming the corrected ADU into temperature or radiance physical units, is not feasible with our setup. The main reason is that our experimental setup does not allow us to maintain a fixed scene radiance while changing the environment and camera temperature. The environment radiance impacts the scene radiance, as the blackbody source emissivity of 0.96 is not uniform.

Figure 10 shows (computed using Equation (Equation 6)) the relative difference of scene radiance caused by the ambient radiance reflected onto the blackbody source surface between a blackbody of emissivity ϵ = 1 and a blackbody with ϵ = 0.96. As the blackbody temperature increases, the influence of ambient radiance at $T_{amb}$ diminishes. Unless the blackbody temperature matches the ambient temperature, the pixel ADU level contains a signal caused by the environment. This limits the achievable accuracy of our calibration experiment (see Section 4).

### 2.8. Acquisition Procedure and Dataset

The camera acquires images of a constant blackbody source radiance continuously, while the climatic chamber (ambient temperature) is heated across a range of predefined temperatures of −5 °C, 0 °C, +5 °C, +10 °C, and +15 °C (see Figure 11). The frame acquisition rate was set at half the camera’s capacity, which is approximately 4 images per second. To probe ambient and flat-field support temperatures, we used two Sensirion STS-35-DIS temperature sensors with an accuracy of ±0.1 °C for the +20 to +60 °C range and ±0.2 °C for the −40 °C to +20 °C range. Data are transferred to the Raspberry Pi board through general-purpose input/output pins. The goal is to emulate imaging operation conditions at the Observatoire de Haute-Provence, where the camera will be installed with the StarDICE instrumentation. Due to time and hardware constraints, including lower or higher temperatures with more setpoints was not feasible. It was not possible to reach blackbody temperatures lower than −30 °C with ambient temperatures above 15 °C as the relative humidity level was too high to avoid water condensation on the device, which could cause malfunction or permanent damage. Each image sequence is repeated for blackbody source temperature setpoints of −30 °C, −20 °C, and −10 °C.

IR images in raw 14-bit format are saved as FITS files containing temperature logs in metadata headers for post-processing (e.g., FPA temperature readings, flat-field support surface temperature, ambient temperature, blackbody temperature). To ease the control command of the camera and other equipment (see Figure 3), we developed multiple Python programs (code for the FLIR Tau2 camera with TeAx ThermalGrabber: https://github.com/Kelian98/tau2_thermalcapture, accessed on 1 January 2024; code for the IR2101 blackbody: https://github.com/Kelian98/ir2101_blackbody, accessed on 1 January 2024).

Finally, to obtain more accurate temperature readings (e.g., FPA temperature, ambient temperature, camera housing temperature), readings were linearly interpolated for each image (taken at 9 Hz), as original sensors temperature measurements were sampled each time an FFC was executed, i.e., at 30 s intervals. Indeed, the camera’s ThermalGrabber interface has an FPGA with two channels (serial for camera control-command and FTDI for raw image data transfer) that cannot be accessed simultaneously. Therefore, images and temperatures cannot be retrieved concurrently.

### 2.9. Data Fitting

The proposed approach hinges on a forward modeling technique for the estimation of the scene radiance. Indeed, we do not have a direct measurement of the true radiance measured by the camera. The forward model computes the scene radiance observed by the instrument $L_{scene}^{inst}$ given by Equation (Equation 6), using $\epsilon_{BB}$, $T_{BB}$, and $T_{amb}$ parameters. Then, the vector of the calibration model parameters $\theta_{i,j}=\left\{g_{i,j},\;o_{i,j},\;\alpha_{i,j},\;\beta_{i,j},\;\gamma_{i,j}\right\}$ of $L_{obs}^{inst}$ defined in Equation (Equation 14) is adjusted onto this scene radiance. These parameters are fitted for each pixel (*i*, *j*) of the FPA with the standard least-squares algorithm by minimizing the independent $\chi_{i,j}^{2}$ likelihood objective functions:(15)$$-2\ln L_{i,j}=\chi_{i,j}^{2}\left(\theta_{i,j}\right)={\Delta L_{i,j}}^{T}\;\cdot \;C_{stat,\;i,j}^{-1}\;\cdot \;\Delta L_{i,j}$$
where the vector of residuals $\Delta L_{i,j}$ describes the differences between the scene radiances $L_{scene}^{inst}$ (computed using Equation (Equation 6)) and the camera response model $L_{obs,\;i,j}^{inst}$.
(16)$$\Delta L_{i,j}=L_{scene}^{inst}\left(\epsilon_{BB},T_{BB},T_{amb}\right)-L_{obs,\;i,j}^{inst}\left(\theta_{i,j},S_{i,j},T_{FPA},T_{cam},T_{amb},T_{FFC}\right)$$
where the best-fit model parameters $\theta_{i,j}$ are those that minimize the $\chi^{2}$ parameter (or maximize the likelihood), $S_{i,j}$ is the camera’s raw response, and $\epsilon_{BB}$ = 0.96 is the blackbody emissivity. $T_{FPA}$ and $T_{cam}$ are vectors of the temperature readings. $C_{stat,\;i,j}$ is the covariance matrix per pixel (i,j) combining uncertainties ($\epsilon_{BB}$, $T_{BB}$, $T_{amb}$) from the scene radiance model detailed in Section 2.2 and the microbolometer readout noise, which is equal to the NERD. We do not consider photon noise as the number of photons received by each pixel during one exposure as $N_{ph}$ is very large for this type of sensor operating in the LWIR band:(17)$$N_{ph}\approx L\times \frac{\pi d^{2}}{4}\times {\left(\frac{p}{f}\right)}^{2}\times \frac{\bar{\lambda }}{hc}\times t_{exp}$$
where *L* is the total radiance perceived by the pixel of size *p* through the lens of aperture diameter *d* and focal length *f* during an exposure time $t_{exp}$ at the instrument effective wavelength $\bar{\lambda }=10.88$ µm. For *L* = 10 $W\;m^{-2}\;{\mathrm{sr}}^{-1}$ and our camera properties, $N_{ph}\approx \;8\times 10^{10}$, resulting in an $SNR=\sigma_{ph}/N_{ph}=\sqrt{N_{phot}}\approx \;2.8\times 10^{5}$. For a radiance equivalent to the NERD or NETD, $SNR\approx \;1.4\times 10^{4}$.

The covariance matrix is diagonal in this case, where variables are assumed to be uncorrelated and with its components $C_{mm}$ defined as
(18)$$C_{mm}=\sigma_{m}^{2}\left(L_{scene}^{inst}\right)$$
where $\sigma_{m}\left(L_{scene}^{inst}\right)$ is the uncertainty in the *m*-th data point. It is computed with the bootstrapping method [39] by drawing random samples from a multivariate normal distribution with $\sigma (\epsilon_{BB})$ = 0.02 (the blackbody surface emissivity uncertainty), $\sigma (T_{BB})$ = 0.1 °C (the blackbody surface temperature uncertainty), and $\sigma (T_{amb})$ = 0.2 °C (the ambient temperature uncertainty). Regressions are performed using the iminuit package [40]. The estimation of the best-fit parameters is conducted with the MIGRAD method.

## 3. Results

### 3.1. Flat-Field Correction

Figure 12 shows a comparison of two images of a blackbody acquired under the same conditions after FFC using the internal shutter and with our custom external flat-field support. The method is the following. We perform an FFC using the internal shutter and directly acquire data from a temperature-stabilized blackbody surface. This operation is then repeated using the custom external support for the FFC. Both resulting images are then subtracted by their respective averages. These operations are carried out one after the other so that the environmental conditions are identical. In using the internal shutter, images show a gradient toward the periphery. This arises from the camera’s reflection, commonly referred to as the Narcissus effect [41], which occurs when infrared radiation emitted by the camera’s internal components, such as the sensor or the lens, reflects back into the sensor, especially against the shutter positioned close to the sensor. This can lead to unwanted artifacts and distortions in the captured images, affecting the accuracy of thermal measurements. In addition, the effect of the lens is not corrected by the internal FFC. With our external FFC procedure, the spatial noise is ∼10 times lower with the spatial standard deviation of the image equal to 4.52 ADU compared to 40.45 ADU after internal FFC. We observed a ten-fold improvement in spatial uniformity by implementing our flat-field support.

### 3.2. Calibration Fit

The calibration matrices are shown in the left column of Figure 13. Correlations between adjacent pixels are obvious for all parameters, and a residual trace of the lens is also visible. Regressions produced high signal-to-noise ratio-fitted parameters. The average $\chi^{2}/dof$ is ∼0.95, indicating a good estimate of the uncertainty on the scene radiance. After applying the calibration model of Equation (Equation 14) with these matrices, all pixels show a consistent behavior when observing scenes with uniform radiance, aligning with the same standard curve. The average root mean square error (RMSE) of the residuals between $L_{scene}^{inst}$ and the fitted $L_{obs}^{inst}$ for all pixels is 0.096 $W\;m^{-2}\;{\mathrm{sr}}^{-1}$.

### 3.3. Spatial Noise

Figure 6 depicts residuals between the estimated scene radiance and one calibrated image. The vignette effect has been canceled on the calibrated image, which appears spatially flat and homogeneous. The fixed-pattern noise (FPN) appears as vertical stripes. The spatial noise (represented by the fitted standard deviation of the Gaussian curve of Figure 6b) is 0.029 $W\;m^{-2}\;{\mathrm{sr}}^{-1}$. It is close to the camera’s NERD = 0.026 $W\;m^{-2}\;{\mathrm{sr}}^{-1}$ for this scene radiance of $L_{scene}$ = 16.313 $W\;m^{-2}\;{\mathrm{sr}}^{-1}$(see Figure 5). Following calibration, the histogram of pixel values closely resembles a Gaussian curve. The error between the average of the calibrated image radiance (given by the Gaussian-fitted mean μ) and the scene effective radiance is −0.018 $W\;m^{-2}\;{\mathrm{sr}}^{-1}$, well within the ±1 −σ range of 0.085 $W\;m^{-2}\;{\mathrm{sr}}^{-1}$ of the scene radiance estimation, computed from the propagation of the sensors’ temperature statistical uncertainties on the forward model.

### 3.4. Application to Sky Radiance Images

To evaluate the efficiency of the calibration model, we compare simulations with radiances of sky images acquired across multiple nights. A series of simulations using libRadTran radiative transfer code [42] were run for three standard atmosphere profiles at a base altitude of 650 m above sea level: mid-latitude summer, winter, and US76 [43]. The air pressure and ground temperature were measured on-site with a dedicated weather station. Ozone concentration and satellite-based PWV measurements were retrieved from the ERA5 dataset [44] using the package cdsapi (https://github.com/ecmwf/cdsapi, accessed on 1 March 2024). All of these parameters were given as input for each simulation, corresponding to one image acquired at a specific time and zenith angle during the night. The framework computes the atmospheric spectral radiance, which was then integrated over the throughput curve shown in Figure 4. Sequences were cautiously selected to be free of apparent clouds and across a sufficient airmass range. We used the DISORT (DIScrete Ordinate Radiative Transfer) method to solve the radiative transfer equation in plane-parallel approximations [42], as zenith angles stayed below 45°. As the full image FOV is large, we focused here on a small fraction that matches the line of sight of the StarDICE optical photometry telescope. The observed radiance was calculated as the mean of a crop area of 32 × 32 pixels near the center of the image, each pixel being close to 1 arcmin across. Figure 14a shows downward spectral radiances computed between 6.5 and 15 µm. Water vapor was uniformly scaled to produce a series of PWV values ranging from 0 (dark red) to 25 mm (blue) with all other atmosphere model parameters held constant: airmass = 1, air pressure = 937 hPa, total ozone column = 300 DU, no aerosol, surface temperature = 273.15 K, and albedo = 0.1. Figure 14b depicts the corresponding radiances integrated over the instrument throughput. They can be used to quantify the impact of changes in PWV on downward radiances for a fixed atmosphere. The integrated scene radiance is intended to simulate our thermal sensor’s calibrated reading. A higher PWV amount leads to higher sky radiances, which can be interpreted as a lowering in altitude of the effective emission level due to increased water vapor opacity, with lower altitudes generally corresponding to higher temperatures [45].

Figure 15 presents a comparison of calibrated radiance data and simulations for three nights of the 2023 northern hemisphere summer (13 September), fall (November 22), and winter (17 December). The default atmosphere profiles of the mid-latitude summer, winter, and US 76 standard models were used as input to the libRadTran simulations with the PWV integral column normalized to the value measured by satellite observations from the ERA5 dataset. The radiance increase with airmass is well reproduced by our calibrated data. The residuals between the calibrated data and a fitted second-order polynomial curve (green) are shown in the lower panels. The small dispersion, with RMSE values below 0.1 $W\;m^{-2}\;{\mathrm{sr}}^{-1}$, indicates excellent smoothness of the data for a clear sky and matches the results of the calibration experiment. The mid-latitude summer model is closest to experimental data with, however, a global offset of 2.5 $W\;m^{-2}\;{\mathrm{sr}}^{-1}$ across the three datasets. This could be partly explained by the presence of a warm foreground. Indeed, the instrument was placed on an equatorial mount inside a dome that radiates toward the camera. The simulated values are systematically below the calibrated observations for all models in Figure 15a (summer), Figure 15b (fall), and Figure 15c (winter), with average PWV values of 24.98 mm, 6.29 mm, and 7.27 mm, respectively. These findings align closely with those of Hack et al. [23], who observed biases toward smaller radiance values in simulations using standard atmosphere profiles. They demonstrated the significance of the vertical distribution of water vapor content with radiosonde profiles taken above their specific site. Measurements of PWV from satellite observations are not precise enough due to the temporal (i.e., 1 h) and spatial horizontal (0.25 × 0.25°) resolutions. Additional instruments such as a GNSS receiver may be required to properly measure the local value of PWV with 1 mm accuracy [46]. Using the local vertical distribution profiles of temperature, water mixing ratio, and other atmospheric parameters should ensure a better adjustment of the simulation to the data.

## 4. Discussion

### 4.1. Bias in Scene Radiance Estimation

The use of a forward model to estimate scene radiance through the use of Equation (Equation 6), while effective within the defined constraints, incorporates simplifications that may not fully capture the intricacies of heat transfer and environmental exchanges. Any deviation from the model’s assumptions can introduce systematic bias. The potential for over- or underestimation under specific environmental conditions unaccounted for in the model raises concerns about the validity of the calibration when using the camera in real-world settings outside the calibration environmental conditions. Low-level sky radiance measurements may suffer from biases. Furthermore, the radiance range of the camera during on-sky measurements differs from that available during calibration. As a result, calibration may need to be extrapolated beyond this range (12–30 $W\;m^{-2}\;{\mathrm{sr}}^{-1}$) for on-sky measurements and may lead to errors when assessing the true-scene radiance. In Section 2.7, we discussed the impact of ambient radiation variation in the laboratory setup. It required us to introduce the additional term fitted with the γ parameter. For on-sky observations, in an open environment, this term will not impact our measurements and is thus not applied for calibrating the associated images. However, in the on-sky observations presented in Figure 15, similar complications may arise due to the emission of the dome hosting the experiment. In an attempt to efficiently deal with this issue, the experiment was recently moved to an entirely open building.

### 4.2. Remaining Fixed-Pattern Noise in Calibrated Images

In Figure 6a, some remaining structures appearing as vertical stripes persist. The dispersion of these residuals (σ=0.0295 $W\;m^{-2}\;{\mathrm{sr}}^{-1}$) is very close to the Noise-Equivalent Radiance Difference (NERD) value (0.0260 $W\;m^{-2}\;{\mathrm{sr}}^{-1}$) provided by the manufacturer. The UIRTC used in this work has read-out electronics similar to those of an optical CMOS-based camera. A shift from a chip-level to column-level analog-to-digital converter (ADC) was made, allowing for lower-speed and lower-power operation. However, this approach introduces column-fixed pattern noise (FPN), compromising image quality. This effect originates from the mismatch between the electronic gain and bias of the ADC channels [47]. These parameters temporally fluctuate. Therefore, the residual structure is always spatially evolving, and the FPN remains visible even after subtracting two consecutive images. Existing complementary image post-processing methods involving deconvolution [48] or neural networks [49] can significantly reduce this residual artifact, but  are beyond the scope of the presented work.

### 4.3. Non-Linearity Correction in Pixel Response

Any detector’s response is not linear for its entire operation regime and deviates at the edges [34]. As a result, the linear approximation is no longer valid as the response saturates at the upper end and stagnates at a non-zero bias level for the lower end. This non-linearity is partially caused by the read-out electronics [34,50]. Multiple expressions for modeling the non-linearity function $\eta \left(S_{i,j}\right)\to S_{i,j}^{\prime }$ have been proposed. Zhou et al. [51] proposed a fixed S-shape curve formulation with the natural logarithm: $S_{i,j}^{\prime }=\ln\left(BD/S_{i,j}-1\right)$, with BD being the bit depth of the sensor ($2^{14}-1$ for our instrument). Lane and Whitenton [34] took the linearity fit equation from Saunders and White [52] with four parameters to adjust (considering the gain *k*): $S_{i,j}^{\prime }=S_{i,j}\times \eta \left(S_{i,j}\right)$ with $\eta \left(S\right)=k\times (1-\left(a+b\times S^{c}\right))$. Other models employing polynomials are also worth studying. If one were interested in higher radiance levels, up to the saturation limit of the sensor, a logistic function model may be preferred. Nevertheless, it is important to note that accurately mapping non-linearity requires measurements across the sensor’s entire sensitivity range, which was not possible with our experimental setup.

### 4.4. Correlation between Scene Radiance and Camera Temperature

The origin of the degeneracy between ambient and FPA temperatures lies in the environmental conditions shared by the sensor and the scene. The ambient temperature influences the scene radiance captured by the camera through reflections on the blackbody source, and it affects the temperature of the camera’s FPA. This co-dependence challenges radiometric calibration, as changes in ambient temperature lead to simultaneous variations in FPA temperature, creating a degenerate relationship between these two variables. Consequently, distinguishing the specific impact of each temperature becomes intricate, with covariances and correlations that complicate the accurate determination of model parameters during the calibration process.

An effective strategy to mitigate this challenge involves placing the camera within a climatic chamber, equipped with an aperture for external exposure. This setup can incorporate moving blackbody sources, akin to the calibration bench used by Lin et al. [53]. Such a configuration offers the potential to disentangle the influence of ambient and FPA temperatures, facilitating more accurate and robust calibration procedures. However, replicating the calibration setup with a climatic chamber and moving blackbody sources proved unfeasible in our study due to limitations in infrastructure, and budget and time constraints.

## 5. Conclusions

In this study, we presented a comprehensive calibration procedure suitable for a UIRTC and aimed at improving its performance to measure the sky downwelling radiance and detect cirrus clouds. A dedicated calibration bench using a climatic chamber and a thermoregulated blackbody source was set up to calibrate the FLIR Tau2 IR thermal camera. A specific analytical calibration model relying on radiance was used. The scene radiance was estimated with a tailored forward model approach. It demonstrated excellent accuracy regarding the average RMSE of 0.1 $W\;m^{-2}\;{\mathrm{sr}}^{-1}$ per pixel and spatial noise roughly equal to the manufacturer’s NERD at ∼0.03 $W\;m^{-2}\;{\mathrm{sr}}^{-1}$. We created matrices of calibration coefficients that can be applied to raw images for on-sky measurements.

We discovered that the surrounding environment is a disturbance to the accuracy of both in-lab and on-sky radiometric measurements as it contributes a radiance that is difficult, if not impossible to model. We think that carefully designed baffling (e.g, made out of aluminium) could significantly reduce this systematic effect.

Applying the following additional adjustments may further improve the calibration experiment and facilitate the analysis: (i) increase the time duration of the climatic chamber temperature set points to let the camera temperature stabilize even more; (ii) use a larger climatic chamber in combination with a larger blackbody emitting surface to increase its distance to the camera and reduce convective and radiative heat transfers between instruments; (iii) consider additional blackbody temperature set-points at lower temperatures; (iv) extend the climatic chamber temperature range; and (v) repeat the measurements multiple times and/or regularly to evaluate repeatability and variations in time.

Additional nights of data collection will occur at the Observatoire de Haute-Provence for the StarDICE experiment. In parallel to IR radiometric measurements, the high-precision optical photometry and spectrophotometry of reference stars are being performed. Future work will focus on establishing a joint analysis between LWIR radiometric images and visible to near-IR photometric data. We hypothesize that through combining on-site and off-site high-accuracy atmosphere sensing (LIDAR [54,55], weather stations, local AERONET data [45,56,57], satellite data [58] and GNSS PWV measurements [46]), it will be possible to (i) improve the modeling of the sky background radiance of LWIR images using libRadTran [42] and isolate cloud structures after PWV emission subtraction [59]; and (ii) quantify the optical photometric flux loss due to atmospheric gray extinction variations. It is still uncertain to which level we can correct the optical photometry with LWIR radiometric measurements. We are currently working on the quantitative assessment of the detection limit of this technique, which will be the subject of a forthcoming paper.

## Figures and Tables

**Figure 1 sensors-24-04498-f001:**
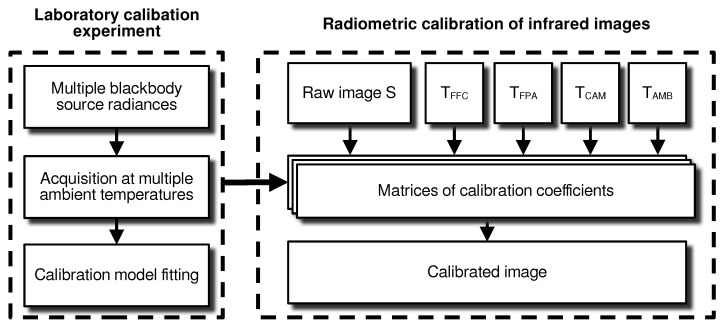
Flowchart of the calibration process represented in two distinct blocks.

**Figure 2 sensors-24-04498-f002:**
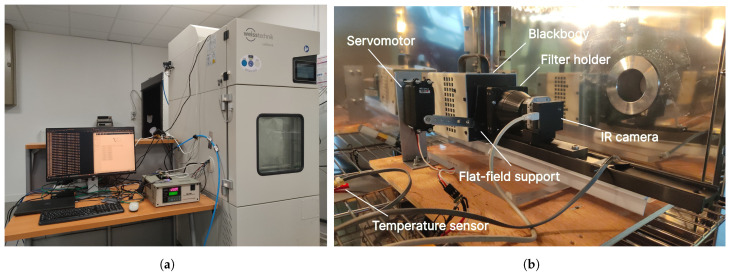
(**a**) Picture of the entire calibration setup with the climatic chamber on the right side of the picture. (**b**) Annotated photo of the interior of the climatic chamber.

**Figure 3 sensors-24-04498-f003:**
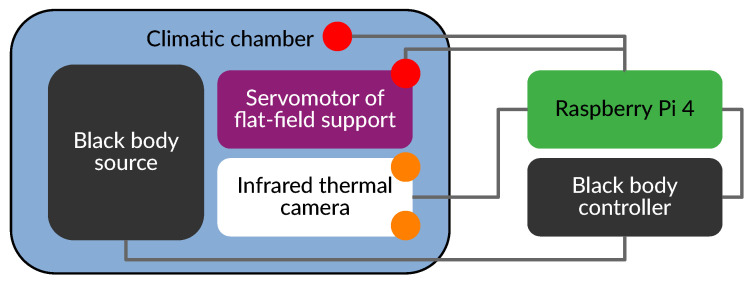
Sketch of the calibration setup bench. The red circles represent additional temperature sensors. The orange circles represent temperature sensors already installed inside the camera.

**Figure 4 sensors-24-04498-f004:**
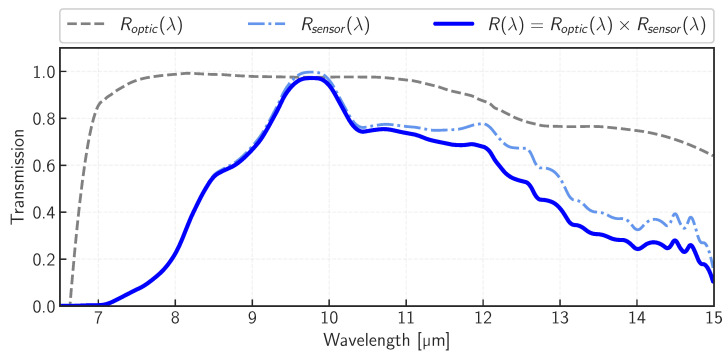
Spectral responses of the FLIR Tau 2 camera core and Umicore F#1.25 f = 60 mm lens.

**Figure 5 sensors-24-04498-f005:**
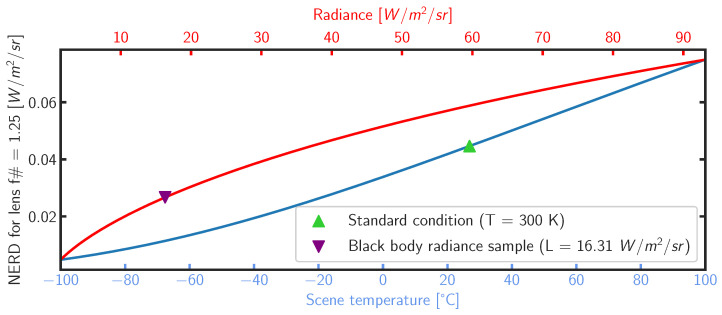
Evolution of the camera NERD as a function of target temperature (blue) and equivalent scene radiance (red) assuming unity emissivity. The y-axis is common for the two curves. The green triangle depicts the NERD = 0.044 $W\;m^{-2}\;{\mathrm{sr}}^{-1}$ in standard operating condition. The purple reversed triangle depicts the expected NERD = 0.026 $W\;m^{-2}\;{\mathrm{sr}}^{-1}$ for the sample image presented in Figure 6.

**Figure 6 sensors-24-04498-f006:**
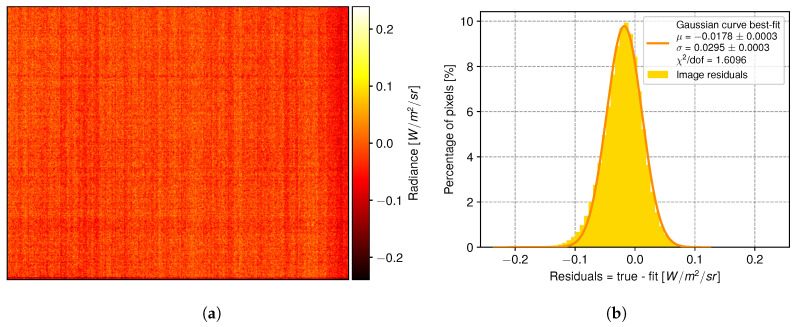
(**a**) Colormap displaying the residuals between a calculated scene radiance and a calibrated blackbody image in $W\;m^{-2}\;{\mathrm{sr}}^{-1}$. (**b**) Histogram distribution of pixel radiance residuals with a Gaussian fit curve.

**Figure 7 sensors-24-04498-f007:**
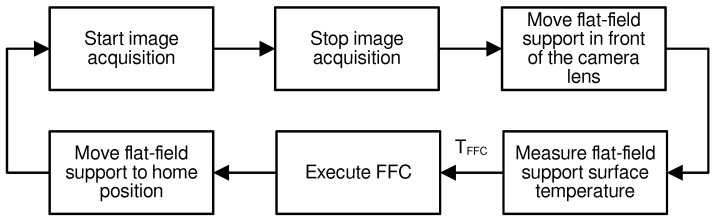
Flowchart of the non-uniformity correction process.

**Figure 8 sensors-24-04498-f008:**
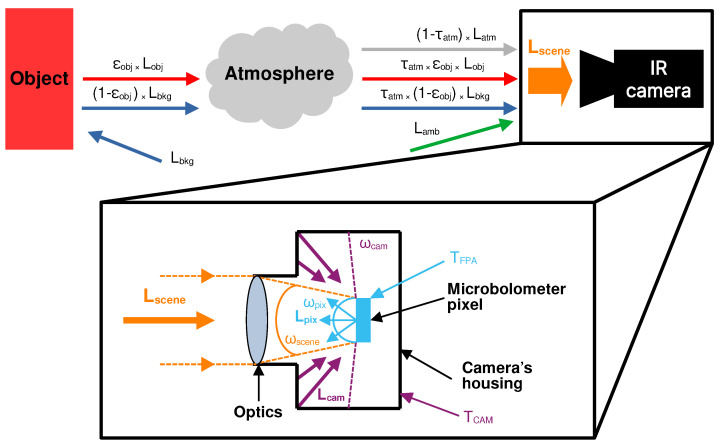
Schematic of radiance contributions for radiometric measurements. The upper panel details the contribution from the outside environment, whereas the lower panel illustrates radiative transfers occurring inside the camera.

**Figure 9 sensors-24-04498-f009:**
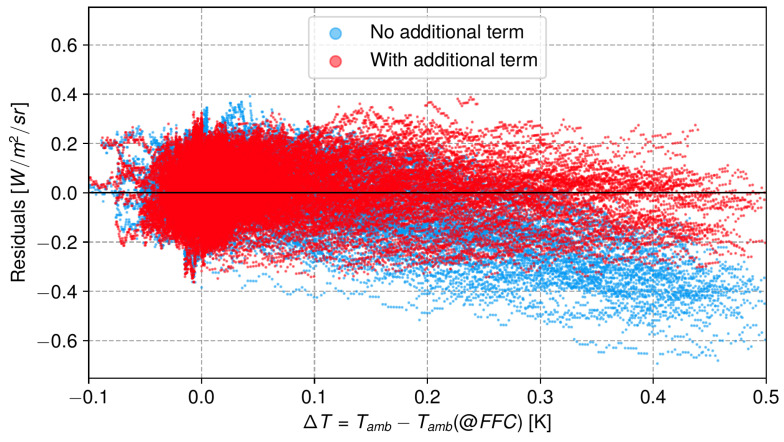
Residuals for a single pixel fitted using the model described in Equation (Equation 8). Blue dots represent the model without any additional terms, while red dots indicate the fit incorporating the γ parameter and the difference $L_{amb}-L_{amb}\left(@FFC\right)$.

**Figure 10 sensors-24-04498-f010:**
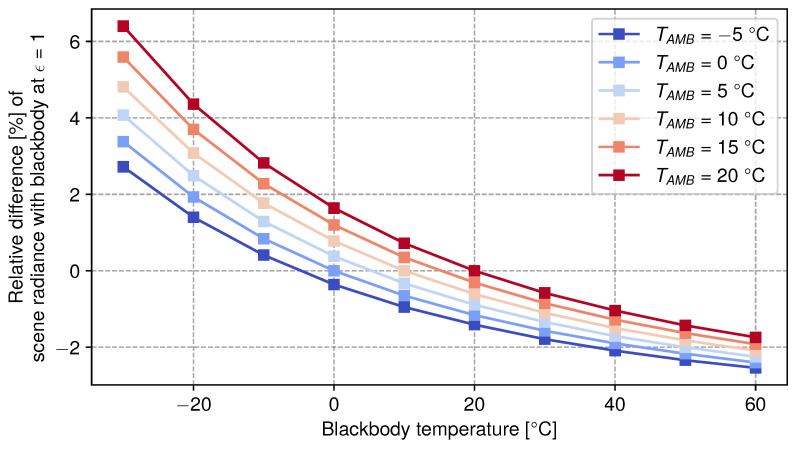
Relative difference of the ambient radiance contribution surrounding the blackbody source with ϵ = 0.96 against perfect blackbody ϵ = 1 during in-lab calibration experiments.

**Figure 11 sensors-24-04498-f011:**
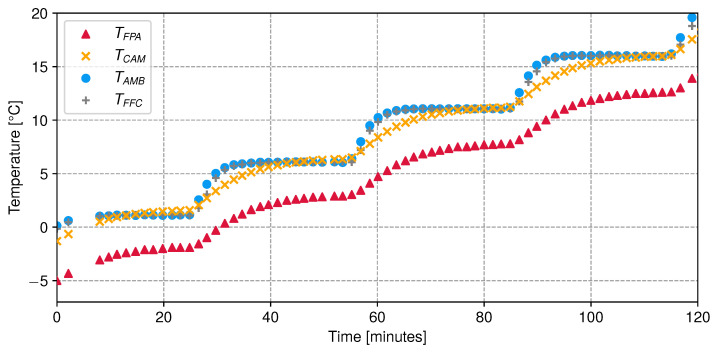
Evolution of temperatures for a blackbody temperature setpoint of −20 °C.

**Figure 12 sensors-24-04498-f012:**
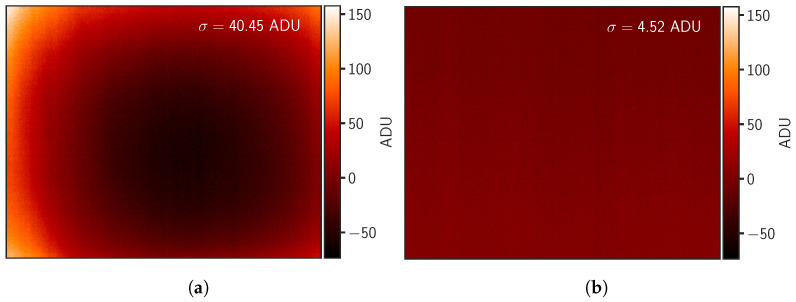
(**a**) Image of a blackbody captured with the default internal camera shutter support for NUC and (**b**) captured with our custom external flat-field support for NUC. Color scales are kept identical for both images. Data consist of mean-subtracted raw images to only show differences in ADU.

**Figure 13 sensors-24-04498-f013:**
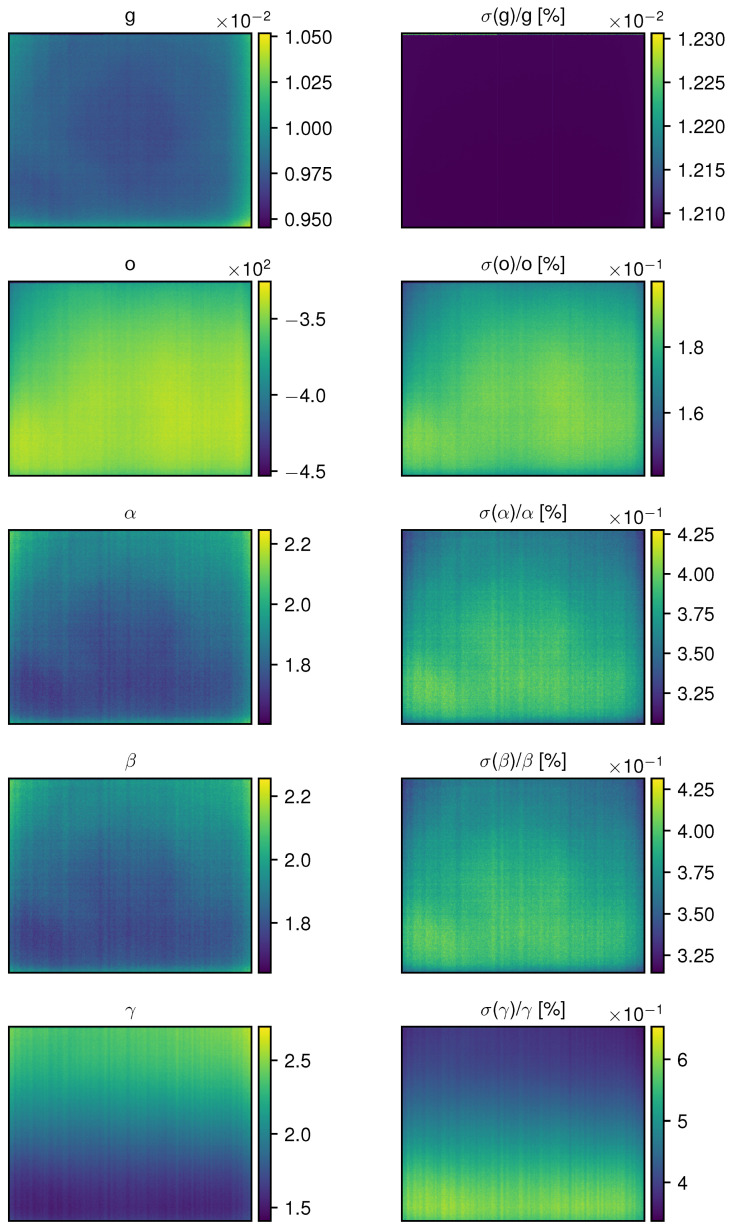
Coefficient matrices reconstructed by fitting the model in Equation (Equation 14) to the calibration data. The right column details the relative uncertainty on the fitted parameters in percentages.

**Figure 14 sensors-24-04498-f014:**
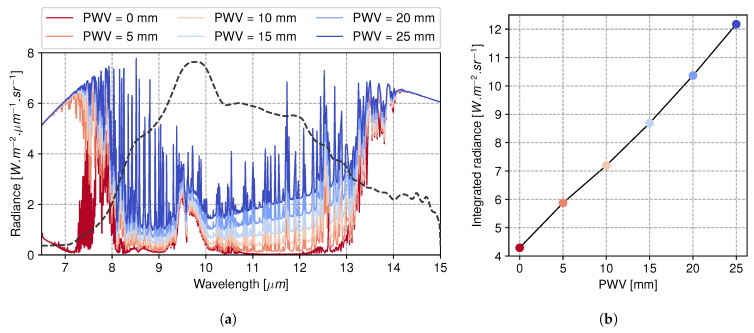
(**a**) libRadtran-simulated spectral downward radiances at zenith of Observatoire de Haute Provence with range of PWV. (**b**) Spectral radiances integrated over instrument throughput (dashed gray line of left panel).

**Figure 15 sensors-24-04498-f015:**
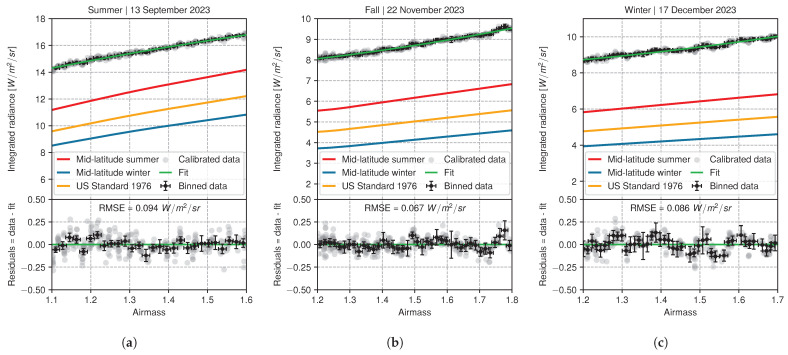
Experimental data compared to libRadTran simulations for three atmospheric profile models and three nights of observation in Summer (**a**), Fall (**b**) and Winter (**c**). The black error bars are experimental data binned over 0.02 airmass. The lower panels show the residuals after a second-order polynomial fit (green curve) of the calibrated data. The RMSE value is computed for the unbinned data relative to the fitted curve.

## Data Availability

The data presented in this study are available upon request from the corresponding author.

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
