# Peer review of "StarDICE II: Calibration of an Uncooled Infrared Thermal Camera for Atmospheric Gray Extinction Characterization"

_sensors, 2024, doi:10.3390/s24144498_

Round 1
Reviewer 1 Report
Comments and Suggestions for Authors
1. When abbreviations first appear, the full name must be provided,such as StarDICE.
2. How much observation accuracy is required for an infrared thermal camera to achieve an instrumental metrology chain with a targeted accuracy of 1 mmag?
3. Can the conventional cooled thermal cameras meet the observation needs of gray extinction? What are the advantages of uncooled infrared thermal camera compared to cooled thermal cameras? Why choose a cameras?
4. What are the improvements of StarDICE II compared to StarDICE?
5. How did other researchers calibrate UIRTC? The author should provide a detailed description.
Author Response
Dear Referees,
Thank you for your thorough reviews of our paper. We appreciate your valuable feedback and have carefully considered all your comments and suggestions that improve the clarity of the paper.
Below, we address your questions and comments.
Best regards,
Kélian SOMMER
========================================================================
- When abbreviations first appear, the full name must be provided,such as StarDICE.
K.S : we now define StarDICE project’s acronym in the beginning of the introduction.
- How much observation accuracy is required for an infrared thermal camera to achieve an instrumental metrology chain with a targeted accuracy of 1 mmag?
K.S : we do not have enough data on the sky to conclude on this subject; however, we think that the mmag level cannot be reached with this camera, maybe the 10 mmag level. The accumulation of nights with StarDICE will allow to gain another factor of ten. Another paper with proper quantitative estimates (for both radiometric and photometric measurements) will be next in the series.
- Can the conventional cooled thermal cameras meet the observation needs of gray extinction? What are the advantages of uncooled infrared thermal camera compared to cooled thermal cameras? Why choose a cameras?
K.S :
- Conventional cooled thermal cameras have the advantage of being stabilized at a constant temperature. Therefore they are not affected by ambient temperature variations, which drastically simplifies their calibration. Generally (at least models commercialized by FLIR), they have a better NETD (~20mK against uncooled plateau at ~40mK). So the spatial noise is divided by a factor of 2. As said in the previous point, we do not know yet if the current calibration of 0.1 W/m2/sr (temporal per pixel) and 0.028 W/m2/sr (spatial per image) is sufficient to reach the mmag level.
- The advantage of the uncooled cameras is their ease of use. They do not require dedicated DC power block and one USB cable is sufficient to use them. They are also far cheaper (~10 times) than their cooled equivalent. They are lighter and easier to integrate onto a telescope mount.
- The IR camera is the solution chosen in an internal LSST collaboration document by Blanc et al. 2013. At this time, this solution appeared to be the easiest to detect any cloud structure in the FOV of the photometric instrument. There may be alternative such as radiometers, LIDARs or RADARs, but they are more expensive and difficult to operate.
- What are the improvements of StarDICE II compared to StarDICE?
K.S : To avoid some confusion:
- StarDICE is the experiment.
- StarDICE I cited in this article refers to the first article of the series published in Astronomy and Astrophysics review : https://www.aanda.org/articles/aa/abs/2023/02/aa44973-22/aa44973-22.html. We cite it here as a reference.
- StarDICE pathfinder is the first version of the experiment to evaluate our ability to build a complete instrumental chain to recalibrate CALSPEC standards with 1 mmag or better.
- StarDICE II is this article, the second in the series, that focus specifically on the calibration of the infrared thermal camera.
If your question concerned the StarDICE pathfinder experiment compared to the actual version of the experiment, a lot of improvements have been made to the different calibration benches to : (i) measure the spectral transmission of the telescope; (ii) transfer NIST absolute calibration reference from photodiode to CCD camera; (iii) build a dedicated calibrated bench to measure radiances of artificial stars; (iv) implement a solution to measure gray extinction → this infrared thermal camera.
- How did other researchers calibrate UIRTC? The author should provide a detailed description.
K.S : in Section 2.1 motivation second paragraph, we explain the standard way to calibrate UIRTCs we found in the literature. We also refer the reader to a well-written comparison review of some techniques from González-Chávez et al. We think this is out of the scope of this paper to provide a detailed review of other techniques.
Reviewer 2 Report
Comments and Suggestions for Authors
The article investigates a new calibration process with a tailored forward modeling approach to achieve accurate measurements with thermal imaging systems. The authors employ a tailored forward modeling approach. Results demonstrate the calibration effects with improved RMSE.
The overall writing of this article is quite good. It is suggested that the discussion section be further organized to make it clearer and more coherent. The conclusion is too detailed. I recommend enhancing it by providing valuable contributions regarding methodologies in this field and suggestions for future development directions.
Comments on the Quality of English Language
This article is overall well written.
Author Response
Dear Referees,
Thank you for your thorough reviews of our paper. We appreciate your valuable feedback and have carefully considered all your comments and suggestions that improve the clarity of the paper.
Below, we address your questions and comments.
Best regards,
Kélian SOMMER
======================================================
The article investigates a new calibration process with a tailored forward modeling approach to achieve accurate measurements with thermal imaging systems. The authors employ a tailored forward modeling approach. Results demonstrate the calibration effects with improved RMSE.
The overall writing of this article is quite good. It is suggested that the discussion section be further organized to make it clearer and more coherent. The conclusion is too detailed. I recommend enhancing it by providing valuable contributions regarding methodologies in this field and suggestions for future development directions.
K.S : we have reworked the conclusion and added a sentence on the future development directions we are following. We also added a statement on the foreground effect. The abstract was therefore also modified.
The discussion is already structured in 4 sub-sections. We are unsure of what the referee is suggesting us to improve.
Reviewer 3 Report
Comments and Suggestions for Authors
The manuscript presents a novel calibration method for an uncooled infrared thermal camera, which is crucial for improving the accuracy of next-generation astronomical surveys. This paper is rich in content with rigorous reasoning and has the potential to be accepted, but some points have to be clarified or fixed before we can proceed, and positive action can be taken.
1 I suggest that authors check the layout formatting of the figure notes to ensure that they all meet the formatting requirements of the journal for final publication.
2 I suggest that the authors provide a brief flowchart to introduce the NUC process by adding a simple flowchart in Section 2.3.
Author Response
Dear Referees,
Thank you for your thorough reviews of our paper. We appreciate your valuable feedback and have carefully considered all your comments and suggestions that improve the clarity of the paper.
Below, we address your questions and comments.
Best regards,
Kélian SOMMER
============================================================
The manuscript presents a novel calibration method for an uncooled infrared thermal camera, which is crucial for improving the accuracy of next-generation astronomical surveys. This paper is rich in content with rigorous reasoning and has the potential to be accepted, but some points have to be clarified or fixed before we can proceed, and positive action can be taken.
1 I suggest that authors check the layout formatting of the figure notes to ensure that they all meet the formatting requirements of the journal for final publication.
K.S: we have shortened the figure captions and included the additional explanation in the text while necessary.
2 I suggest that the authors provide a brief flowchart to introduce the NUC process by adding a simple flowchart in Section 2.3.
K.S: we asked the manufacturer to provide information on the NUC/FFC process, but they would not want to answer for intellectual private property reasons. So we give a simple NUC process summarizing the operation undertaken on our side.
Reviewer 4 Report
Comments and Suggestions for Authors
The paper is in general clear, and results well presented. Anyway some improvements are needed before publishing.
As a general comment, I suggest to stress more on the improvements in the results obtained by the calibration procedure.
- Page 8 line 256; English language: I noticed only a typo at: after TBB it should be a comma not a dot.
- Page 12 lines 359 – 361; You wrote: “Finally, to obtain more accurate temperature readings (e.g., FPA temperature, ambient temperature, camera housing temperature), readings were linearly interpolated for each image (taken at 9 Hz) as original measurements were sampled at 30-second intervals”. Please can you better justify the reason why interpolated data are used instead of acquiring data with a different frequency?
- Page 13 lines 378 – 381; You state: “No error uncertainty in the flux (photon or shot noise) is considered, as the number of photons received by each microbolometer is very large for this type of sensor operating in the LWIR band.” Please justify the statement with calculation and figures for photon noise, shot noise and received signal.
- Page 17 lines 458 – 461; You state: “The simulated values are systematically below the calibrated observations envelope for all models in Fig. 14a (summer), 14b (fall) and 14c (winter), with average PWV values of 24.98 mm, 6.29 mm and 7.27”. Can you explain the reason of such behavior?
- Page 199 lines 536 – 539; At the end of the paper you wrote: “Such a configuration offers the potential to disentangle the influence of ambient and FPA temperatures, facilitating more accurate and robust calibration procedures. However, replicating the calibration setup with a climatic chamber and moving blackbody sources proved unfeasible in our study due to limitations in infrastructure, and budget and time constraints.” Are the authors planning to perform some measurements to disentangle the two effects?
Author Response
Dear Referees,
Thank you for your thorough reviews of our paper. We appreciate your valuable feedback and have carefully considered all your comments and suggestions that improve the clarity of the paper.
Below, we address your questions and comments.
Best regards,
Kélian SOMMER
============================================================
The paper is in general clear, and results well presented. Anyway some improvements are needed before publishing.
As a general comment, I suggest to stress more on the improvements in the results obtained by the calibration procedure.
K.S : we have reworked the conclusion and added a sentence on the future development directions we are following.
- Page 8 line 256; English language: I noticed only a typo at: after TBB it should be a comma not a dot.
K.S : done.
- Page 12 lines 359 – 361; You wrote: “Finally, to obtain more accurate temperature readings (e.g., FPA temperature, ambient temperature, camera housing temperature), readings were linearly interpolated for each image (taken at 9 Hz) as original measurements were sampled at 30-second intervals”. Please can you better justify the reason why interpolated data are used instead of acquiring data with a different frequency?
K.S : during the acquisition, sensors temperatures were read every 30 seconds, when the FFC was performed. The code was originally written sequentially and it was not possible to sample the temperature at the same rate as the imaging frequency of 9Hz. Moreover, there are two different interfaces for controlling the camera (through standard serial communication) and getting images (through FPGA with FTDI communication) and they cannot be used simultaneously. I added additional explanation in the text :
Finally, to obtain more accurate temperature readings (e.g., FPA temperature, ambient temperature, camera housing temperature), readings were linearly interpolated for each image (taken at 9 Hz) as original \ks{sensors temperatures} measurements were sampled \ks{each time a FFC was executed}, i.e. at 30-second intervals. \ks{Indeed, the camera's ThermalGrabber interface has an FPGA with two channels (serial for camera control-command and FTDI for raw image data transfer) that cannot be accessed simultaneously. Therefore, images and FPA or housing camera temperatures cannot be retrieved concurrently.
- Page 13 lines 378 – 381; You state: “No error uncertainty in the flux (photon or shot noise) is considered, as the number of photons received by each microbolometer is very large for this type of sensor operating in the LWIR band.” Please justify the statement with calculation and figures for photon noise, shot noise and received signal.
K.S : we added the equation (17) and a computation in the text.
- Page 17 lines 458 – 461; You state: “The simulated values are systematically below the calibrated observations envelope for all models in Fig. 14a (summer), 14b (fall) and 14c (winter), with average PWV values of 24.98 mm, 6.29 mm and 7.27”. Can you explain the reason of such behavior?
K.S : we rephrased this paragraph, hoping the explanation is now clearer to the reader.
- Page 199 lines 536 – 539; At the end of the paper you wrote: “Such a configuration offers the potential to disentangle the influence of ambient and FPA temperatures, facilitating more accurate and robust calibration procedures. However, replicating the calibration setup with a climatic chamber and moving blackbody sources proved unfeasible in our study due to limitations in infrastructure, and budget and time constraints.” Are the authors planning to perform some measurements to disentangle the two effects?
K.S : we do not plan to do this in the near future due to the lack of infrastructure and finances.